# Microfluidic Microcirculation Mimetic as a Tool for the Study of Rheological Characteristics of Red Blood Cells in Patients with Sickle Cell Anemia

Marcus Inyama Asuquo [1,*], Emmanuel Effa [2], Oluwabukola Gbotosho [3], Akaninyene Otu [2], Nicole Toepfner [4], Soter Ameh [5], Sruti-Prathivadhi Bhayankaram [6], Noah Zetocha [6], Chisom Nwakama [6], William Egbe [7], Jochen Guck [8] and Andrew Ekpenyong [6]

1 Department of Hematology, Faculty of Medicine and Dentistry, University of Calabar, Calabar 540001, Nigeria
2 Department of Medicine, Faculty of Medicine and Dentistry, University of Calabar, Calabar 540001, Nigeria; emma.effa@gmail.com (E.E.); akanotu@yahoo.com (A.O.)
3 Vascular Medicine Institute, University of Pittsburgh, Pittsburgh, PA 15261, USA; bukolagbotosho@yahoo.com
4 Department of Pediatric Hemato-Oncology, Carl Gustav Carus University Hospital, Technische Universität Dresden, 01307 Dresden, Germany; nicole.toepfner@outlook.com
5 Biostatistics Unit, Department of Community Medicine, University of Calabar, Calabar 540001, Nigeria; soterameh@yahoo.com
6 Department of Physics, Creighton University, Omaha, NE 68178, USA; sruti.prathivadhi@gmail.com (S.-P.B.); noahzetocha@creighton.edu (N.Z.); chisom.nwakama@gmail.com (C.N.); andrewekpenyong@creighton.edu (A.E.)
7 Joseph Ukpo Hospitals and Research Institutes (JUHRI), Afua Site, Ibiono Ibom 520115, Nigeria; mathwillis2@gmail.com
8 Biotechnology Centre, Technische Universität Dresden, 01307 Dresden, Germany; jochen.guck@mpl.mpg.de
* Correspondence: marcus.inyama@npmcn.edu.ng; Tel.: +234-8032235333

**Featured Application: Feasibility of personalized medical care for patients with sickle cell anemia using a lab-on-chip microfluidic platform to mimic and unravel components of vaso-occlusive crisis.**

**Abstract:** Sickle cell disorder (SCD) is a multisystem disease with heterogeneous phenotypes. Although all patients have the mutated hemoglobin (Hb) in the SS phenotype, the severity and frequency of complications are variable. When exposed to low oxygen tension, the Hb molecule becomes dense and forms tactoids, which lead to the peculiar sickled shapes of the affected red blood cells, giving the disorder its name. This sickle cell morphology is responsible for the profound and widespread pathologies associated with this disorder, such as vaso-occlusive crisis (VOC). How much of the clinical manifestation is due to sickled erythrocytes and what is due to the relative contributions of other elements in the blood, especially in the microcapillary circulation, is usually not visualized and quantified for each patient during clinical management. Here, we used a microfluidic microcirculation mimetic (MMM), which has 187 capillary-like constrictions, to impose deformations on erythrocytes of 25 SCD patients, visualizing and characterizing the morpho-rheological properties of the cells in normoxic, hypoxic (using sodium meta-bisulfite) and treatment conditions (using hydroxyurea). The MMM enabled a patient-specific quantification of shape descriptors (circularity and roundness) and transit time through the capillary constrictions, which are readouts for morpho-rheological properties implicated in VOC. Transit times varied significantly ($p < 0.001$) between patients. Our results demonstrate the feasibility of microfluidics-based monitoring of individual patients for personalized care in the context of SCD complications such as VOC, even in resource-constrained settings.

**Keywords:** sickle cell disorder; vaso-occlusive crisis; hydroxyurea; microcirculation; microfluidics; personalized medicine; deformation; transit time

## 1. Introduction

Sickle cell disease (SCD) is the most common severe inherited red blood cell disorder, affecting millions of people worldwide [1]. Three-quarters of sickle cell cases in the world occur in Africa with an annual incidence of about 250,000–400,000, mainly in sub-Saharan Africa [2,3]. For example, a WHO report estimated that around 2% of neonates in Nigeria are affected by SCD, giving a total of 150,000 births per year in Nigeria alone [2]. Similarly, in some regions of Africa, more than a quarter of the population carries the SS mutation [2]. The frequency of the sickle cell trait ranges between 15% and 30% in Ghana and Nigeria and from 10% to 40% across equatorial Africa, reaching up to 45% among the Baamba tribe in Uganda and decreasing to 1–2% on the North African coast and less than 1% in South Africa [2]. According to the United States Centers for Disease Control and Prevention, 7.7% (1 in 13) of Black or African American babies are born with the trait, and SCD affects approximately 100,000 Americans [4]. Death from SCD complications occurs mostly in children under five years in Africa, due to the late detection of the condition, inadequate facilities, scarcity of trained personnel, lack of adequate diagnostic tools and insufficient medical care [2].

Sickle Hb (HbS) is due to a mutation resulting in the substitution of thymine for adenine (GAG to GTG) in the sixth codon of the hemoglobin $\beta$ chain gene. This causes the coding of valine (neutral charge) instead of glutamic acid (negative charge) at position six on the $\beta$ chain [5,6]. The alteration in the charge of the HbS molecule accounts for its different mobility abilities under electrophoresis. The loss of a negative charge at this site on the outside of the HbS molecule (cf. normal adult HbA) also enables neighboring molecules to aggregate upon deoxygenation, eventually forming long polymers [7]. Individuals homozygous for HbS (cf. HbA) develop the disease, while heterozygotes (HbAS, or the so-called sickle cell trait), usually lack symptoms, although their RBCs can be induced by sickle ex vivo [8,9]. In addition to homozygote HbSS disease (sickle cell anemia, SCA), seven other major sickle genotypes have been linked to the disease. These include HbSC, HbS/$\alpha$-thalassemia, HbS/$\beta°$-thalassemia, HbS/$\beta$+-thalassemia, HbS/hereditary persistence of fetal Hb (S/HPFH), HbS/HbE syndrome, HbS/G6PD (glucose-6-phosphate dehydrogenase deficiency) and many other rare combinations, such as HbS with HbD Los Angeles, HbO Arab, G-Philadelphia, etc.

In this study, we focus on the feasibility of deploying a cutting-edge microfluidic lab-on-chip device in a low-resource setting to measure prognosis and guide the clinical management of SCD using results of morphological and rheological properties of red blood cells as they traverse the microvasculature under the effect of clinical stress conditions. Our lab-on-chip device, called the microfluidic microcirculation mimetic (MMM), [10–14] mimics both the pulmonary and peripheral microcirculatory environments, measures cell transit times and deformability and assesses the obstructive effects of constrictions on the cells. These measurements and assessments constitute readouts of the mechanical or rheological characteristics of the cells, which, in a broader context, have been shown to have important physiological roles [15–17] and are in the diagnostic focus for characterizing pathologies [18–20]. The main objective of this study was to show the feasibility of using an advanced lab-on-chip microfluidic platform (here, MMM) to unravel the contributions of morpho-mechanical properties of single RBCs to a vaso-occlusive crisis, VOC, to characterize and advance clinical management of SCD complications in individual patients, even in a resource-poor setting. Furthermore, this study points out that it is also possible to use the MMM technique to assess the effects of novel interventions, which are under development, on the microcirculatory vasculature. Of course, this possibility fits with the knowledge that SCD is not merely a monogenic hemoglobin disorder resulting in increased RBC stiffness which leads to VOC but is also a vasculopathic disease. Our work is a necessary preparation for the introduction of more lab-on-chip platforms that will address several old and new questions about SCD complications such as the role of adhesion versus stiffness of cells [21]. The real-time deformability cytometer (RT-DC) [22] and the integrated automated particle-tracking microfluidic [23] platforms are among those whose utility the

present feasibility report assesses for future work. The scientific reasons for an amplified armamentarium emerge from the pathophysiology of SCD and current treatment strategies. Thus, we have established a multinational collaboration involving researchers based in the USA and Germany, who developed advanced cutting-edge tools to directly address clinical questions around SCD management, and clinical researchers in sub-Saharan Africa where SCD burden is the highest for the quick deployment of diagnostic tools with an effective synergy in bench-to-bedside translational research.

Sickling of RBC in the human vascular beds can result in ischemic tissue injury with organ dysfunction and early death [9]. Patients with SCD can develop very painful, life-threatening complications, including extensive organ damage, which can reduce their quality of life and life expectancy. Chronic organ damage involving the central nervous system, the heart, the lungs and the kidneys has also been observed to be part of SCD complications. The complications that follow an acute sickle cell crisis are often infarction or thromboembolic events in any of the organs. Other complications include sensory-neural hearing loss and proliferative and non-proliferative retinopathy [24]. Yet, other complications include splenic fibrosis, autosplenectomy from the repeated episodes of infarction and consecutive immunosuppression. Priapism and chronic leg ulcers are common complications among adolescents, especially those with homozygote sickle cell disease. Still, other complications are nephrotic syndrome, gnathopathy, avascular necrosis of the head of the femur, septic arthritis and osteomyelitis [25]. Patients with SCD have a markedly decreased life expectancy, and their quality of life is greatly compromised by frequent hospitalization caused by the disease [12,21]. The effects of the disease are more marked in children than in adults, and patients die of opportunistic infections (young children and neonates with SCD are especially at risk for VOD during episodes of fever, infections and dehydration. Elderly patients with SCD splenic fibrosis often die of infections caused by encapsulated bacteria such as pneumococci) [19,20].

Treatment of the disease condition depends on the presenting features. Patients presenting with acute events may require emergency care. The most common acute event in SCD patients requiring admission to the emergency department in Nigeria is a painful vaso-occlusive crisis (VOC) [26,27]. Therefore, VOC is the most common clinical challenge addressed in this feasibility study. Treatment in such cases of VOC is a combination of pain relief, fluid therapy and identifying the trigger and treating it [24]. The patients presenting with other crises are treated based on the standards of care. A steady-state treatment is usually divided into treatment of ongoing problems and prevention based on the standards of care. The recurrent and unpredictable nature of VOC episodes [28] makes these treatments and standards of care very limited if not sporadic in successful outcomes.

## 2. Materials and Methods

### 2.1. Setting, Ethical Approval, Participants, Informed Consent and Sample Collection

**Setting:** This study was conducted in the Department of Hematology of the University of Calabar Teaching Hospital (UCTH) Calabar, Cross River State, Nigeria. The UCTH is the only tertiary health institution in the state. It has a 400-bed capacity, a modern ICU and accident and emergency departments. Calabar is the capital of Cross River State and the largest city in the state. The population is about 3.8 million.

**Ethical Approval and Informed Consent:** This study received ethical approval (UCTH/HREC/33/498) from the University of Calabar Teaching Hospital Health Research Ethics Committee. Participants recruited for the study had the study thoroughly explained to them in a language they understand. Those with a poor understanding of English language had interpreters explain the procedure before signing consent forms.

Contents of the form were again explained, and where the subject agreed, they were required to sign or use their thumb print, as shown in Supplementary Information (SI) I.

**Recruitment of Participants**: Patients with sickle cell anemia (SCA) registered at the Haematology Clinic, Calabar Sickle Cell Club or the Paediatric Sickle Cell Clinic, were

recruited to participate in the study, based on the inclusion criteria listed in SI III. All patients who fell within the exclusion criteria were not recruited.

**Questionnaire and Medical History**: Structured questionnaire to capture personal data, demographic, past and present records of sickle cell crisis and complications (SI II) were administered to the participants by trained research assistants.

**Sample Collection:** Venipuncture was used for collection, and trained phlebotomists took 5 mL of blood from the ante cubital fossa of the participants. Blood samples were deposited into an EDTA sample bottle. There was only single sampling of participants' blood.

### 2.2. Microfluidic Microcirculation Mimetic

**Microfluidic Microcirculation Mimetic:** We developed the MMM to mimic the advection of cells in the human capillaries, which often have constrictions smaller than blood cell diameters (Figure 1). To simulate VOC ex vivo, we induced hypoxic conditions that lead to VOC using sodium metabisulfite (SM), a reducing agent which promotes sickling [29]. We also mimicked treatment conditions by treating sRBC of patients with hydroxyurea (HU), an FDA-approved drug whose clinical benefit varies between patients and whose mechanism of action in the case of SCD is not completely understood [25,30]. The MMM differs from existing microfluidic devices and point-of-care technologies [31–33] that model blood vessels in the lungs and measure deformability in that it does not involve branches or channel networks but remains serial. We have maintained a serial model for two reasons. Firstly, from the "viewpoint" of the cell, constrictions occur one after the other, branches or no branches. Secondly, a serial model enables a very large number of constrictions (here, 187) within the field of view of the microscope objective, allowing real-time observation of physiologically or pathophysiologically relevant phenomena such as vaso-occlusion. Detailed descriptions of the MMM can be found in previous work where it has been used to study various cell types in health and diseases, including macrophages [16], neutrophils [11,12], leukemia cells [13] and mesenchymal stromal cells [14]. Briefly, following conceptualization, AutoCAD software was used to design three variants of the MMM: one without constrictions (constant width of 15 μm) and the others with 5 and 7 μm as the smallest constriction widths. All variants have a constant height of 15 μm. A polyester photomask of the design was printed commercially (Photo Data and J.D. Photo-Tools, JD Photo Data, Hitchin, UK, and more recently by Potomac Photonics, Halethorpe, MD, USA). Using the photomask, the master molds were fabricated at the Institute of Semiconductors and Microsystems of TU Dresden, Germany, and at Potomac Photonics, Halethorpe, Maryland, USA, following standard photolithographic techniques. These master molds can then be used hundreds of times to make the MMM chips when needed, using soft lithography [11–13].

For soft lithography (currently done in Creighton University, Omaha, NE, USA), PDMS (Sylgard 184, Dow Corning, Midland, TX, USA) is degassed by centrifugation (1200 rpm for 20 min), poured onto molds and baked for about 20 min at 100 °C. Inlet and outlet holes are punched (Harris Uni-core, 1.5 mm hole) on the cut and peeled PDMS chip. The chips are bonded to 22 mm glass cover slips (Nr 1, Marienfeld, Germany) using oxygen plasma (Plasma Cleaner PDC-32G, Harrick Plasma). The inlet of the MMM chip is connected to a programmable double-channel microfluidics syringe pump (NE-4002X, New Era, New York, NY, USA) using flexible polycarbonate tubing and 1 mL syringe containing about 1000 μL of cell suspension, for each experiment (Figure 1). Cells are advected sequentially through the microchannel using volume flow rates (0.9, 9 and 99 μL/h) that include the physiological blood flow range. For imaging, the device was mounted on a phase-contrast inverted microscope (Amscope, Ningbo, China) with 20× and 40× magnification objectives for measuring cell size and 4–5× objective for measuring the transit time through the entire device.

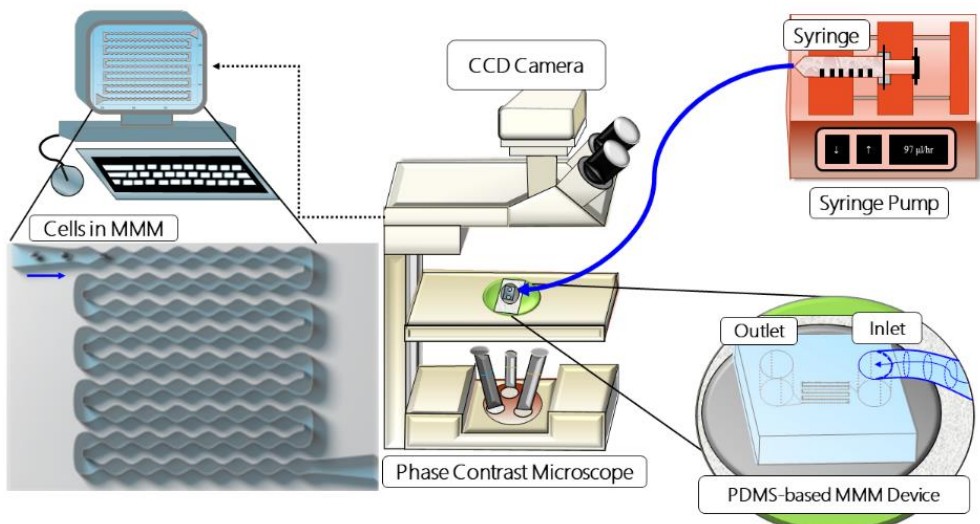

**Figure 1.** Schematic set-up of the microfluidic microcirculation mimetic (MMM). A suspension of PBS-diluted cells in a syringe is fixed to a syringe pump, connected by tubing to the PDMS-based MMM chip and placed on a phase-contrast microscope equipped with a CCD camera. The camera is connected to a computer which enables the monitoring and running of the device, data collection and analysis.

A CCD camera (DMx 21BF04, Germany, Imaging Source, Charlotte, NC, USA) was used to record advection videos at a frame rate of 60 frames per second. This relatively low frame rate is sufficient since transit times through the entire device is at least an order of magnitude longer than the frame interval of 0.0167 s. Even a much cheaper microscope and CCD camera with a frame rate of 15 frames per second (Amscope, Ningbo, China, XSB-1A), which we deployed for most of the reported experiments in a resource-challenged setting, worked well. Transit times were extracted from the videos via off-line analysis. Statistical analyses were carried out using Origin (OriginLab, Northampton, MA, USA).

**Sample Preparation.** The collected blood sample from each participant was diluted with PBS to obtain a final RBC density of about $1 \times 10^6$ cells/mL, for optimal single-cell advection through MMM. Hydroxyurea, HU (from 500 mg capsules, Bristol-Myer, New York, NY, USA, Squibb), was added to a final concentration of 0.4 mg/mL. Sodium metabisulfite, SM (Burgoyne, India, Product # 03491, and Batch # 31077) was added to a final concentration of 2% *v/v*. Dilutions were guided by full blood counts carried out with a Haematology Analyzer (Erma PCE 210, Erma Incorporated, Tokyo, Japan) and confirmed manually via microscopy-based hemocytometer counting. Supplementary Information (SI) Table S1 shows anonymized patient data, including full blood counts from the Haematology Analyzer.

### 2.3. Morphometry during MMM Measurements

We have used several shape descriptors to parameterize and quantify the morphology of red blood cells in the outlet of the MMM (this can also be performed for cells in the inlet). The main parameters are circularity (*C*) and roundness (*R*), defined as follows.

$$C = (4\pi A)/(P^2), \tag{1}$$

where *A* is the area and *P* is the perimeter of the object (cell), ranging from 0 for infinitely elongated polygon (capturing extreme sickling) to 1 for perfect circle (capturing the healthiest RBC, see SI Figure S1).

$$R = (4A)/(\pi \times (2a)^2), \tag{2}$$

where *2a* is the major axis of the object (cell). Obviously, *R* compares the bare dimensions of area and axis ratio (simple amount of elongation of RBCs) while *C* picks up on details of the

roughness of edges based on the perimeter. Morphometry of RBC images from MMM outlet was performed using the software Image J [34]. As shown in SI Figure S1, the cells in the original image were detected (2), converted to gray scale (3) and then binary formatted (4) for particle analysis (5), in which cells on the edges were excluded, size range was applied to exclude debris and particles that were not cells and cells were aggregated, leading to results only for single separate cells (6). The robustness of our single-cell morphometric is evident in SI Figure S1 where, for instance, cells 9 and 14 in panel 6 (Results) are clearly normal and sickled, respectively, as in the original image (panel 1) and have $C = 0.92 \pm 0.02$ and $C = 0.79 \pm 0.02$, $R = 0.95 \pm 0.02$ and $R = 0.64 \pm 0.01$.

### 3. Results

*3.1. Morpho-Rheological Properties of Sickle RBCs Are Patient-Specific*

3.1.1. Transit Times through Microcapillary-like Constrictions

The transit time (*tt*) through the 187 deformation-imposing capillary-like constrictions in the MMM is a readout for the stiffness of cells. We measured the *tt* for 25 prepared blood samples of SCD patients who participated in the study, using three flow rates: 0.9, 9 and 99 µL/h. The lowest flow rate, 0.9 µL/h, enabled the most reliable serial advection of RBCs amenable to *tt* measurement. As shown in Figure 2, we found significant variation in *tt* between patients. Notably, data for 4 patients out of the 25 patients measured are shown in Figure 2 because these were the most comparable measurements in terms of sustained smooth flow through the device throughout the duration of measurement in the MMM (about 15 min).

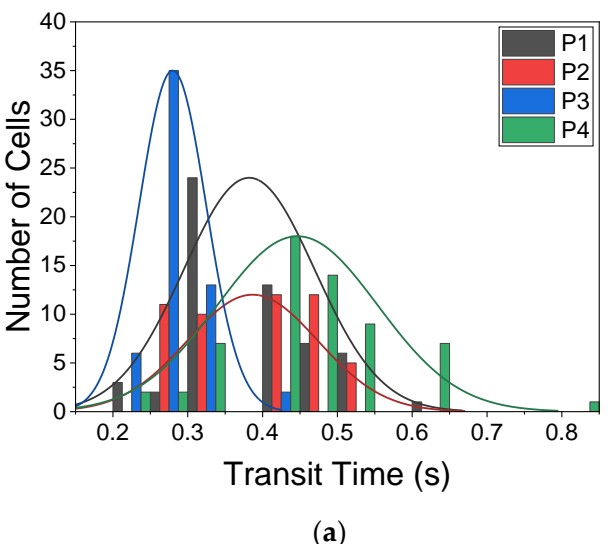
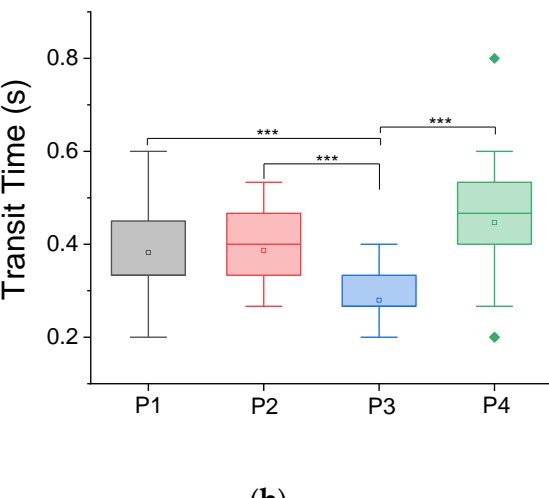

(**a**)                      (**b**)

**Figure 2.** Transit times of RBCs in MMM vary between patients. (**a**) Histogram distribution of transit times from four patients, labeled P1, P2, P3 and P4, with normal (Gaussian) fits. The cell numbers (*n*) and mean transit times (*tt*) at a flow rate of 0.9 µL/h are: P1, $n = 56$, $tt = 0.38 \pm 0.01$ s; P2, $n = 50$, $tt = 0.39 \pm 0.01$ s; P3, $n = 56$, $tt = 0.28 \pm 0.01$ s; and P4, $n = 60$, $tt = 0.45 \pm 0.01$ s. The 5 µm MMM device was used. All given uncertainties are standard errors of the mean. (**b**) Box plots from one-way ANOVA for the transit times data in (**a**). At *** $p < 0.001$, there is a statistically significant difference in *tt* of RBCs from the four patients.

For the four patients shown, denoted as P1, P2, P3 and P4 (Figure 2), cell numbers (*n*) analyzed and the mean *tt* at a flow rate of 0.9 µL/h were: P1, $n = 56$, $tt = 0.38 \pm 0.01$ s; P2, $n = 50$, $tt = 0.39 \pm 0.01$ s; P3, $n = 56$, $tt = 0.28 \pm 0.01$ s; and P4, $n = 60$, $tt = 0.45 \pm 0.01$ s. All given uncertainties are standard errors of the mean. The histogram distribution of transit times or RBCs from these four patients and the normal or Gaussian fits to these histograms illustrate a wide range of differences between patients. To quantify these differences, we carried out analysis of variance (ANOVA), specifically one-way ANOVA, and found that at

*** $p < 0.001$, there is a statistically significant difference in the *tt* of RBCs among the four patients, signaling the need for patient-specific monitoring rather than some fixed expected pathological values.

### 3.1.2. Morphometry Post-Microcapillary-like Constrictions

In the cascade of events that trigger VOC, polymerization of abnormal hemoglobin HbS, leading to sickling, is a key step. To dynamically determine and quantify the extent of sickling and possible contribution to VOC, we measured the circularity, *C*, Equation (1) and roundness, *R*, Equation (2), of RBCs at the exit channel of the MMM following advection. Figure 3 shows the results for circularity, and Figure 4 shows the results for roundness in one patient, patient 4 (P4). The heterogeneity of RBC morphology in the same patient is obvious from Figure 3a, where the histogram of steady-state circularity, P4, has two peaks as made clearer by the kernel smooth curve. The higher peak corresponds to normal RBCs and the lower to sickled RBCs, with a broad range that shows varying degrees of sickling. The distribution was altered when cells were treated with sodium metabisulfite (SM) to induce sickling, as seen in the P4 + SM result of Figure 3a. Finally, when the cells were treated with hydroxyurea (HU), the distribution again departed from the steady-state situation as seen in the P4 + HU result of Figure 3a. Box plots for the circularity data are shown in Figure 3b with statistically significant differences found among P4, P4 + SM and P4 + HU. Since we recorded videos of the MMM at various times post-advection, we could affirm the dynamic nature of morphological heterogeneity within the same patient, even for a specific condition such as P4 + SM or P4 + HU.

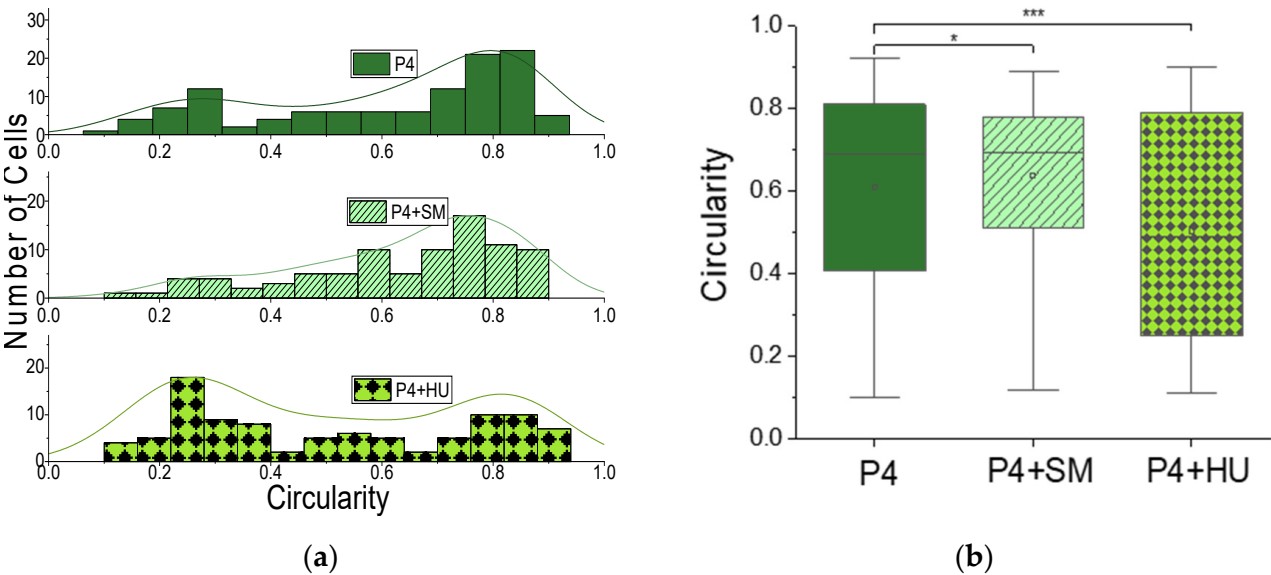

(**a**)            (**b**)

**Figure 3.** Morphometry of cells from MMM outlet: circularity. (**a**) Histogram of circularity (Equation (1)) and kernel smooth curves for RBCs from patient 4, showing morphological heterogeneity in untreated cells, P4; in cells treated with sodium metabisulfite, P4 + SM; and cells treated with hydroxyurea, P4 + HU. (**b**) Box plots for the circularity data in (**a**). Cell numbers (*n*) and circularity (*C*) for each condition are: P4, $n = 114$, $C = 0.61 \pm 0.02$; P4 + SM ($n = 88$, $C = 0.64 \pm 0.02$); and P4 + HU ($n = 9$, $C = 0.50 \pm 0.03$). One-way ANOVA shows statistically significant (*** $p < 0.001$) differences between P4 and P4 + HU and (* $p < 0.05$) between P4 and P4 + SM.

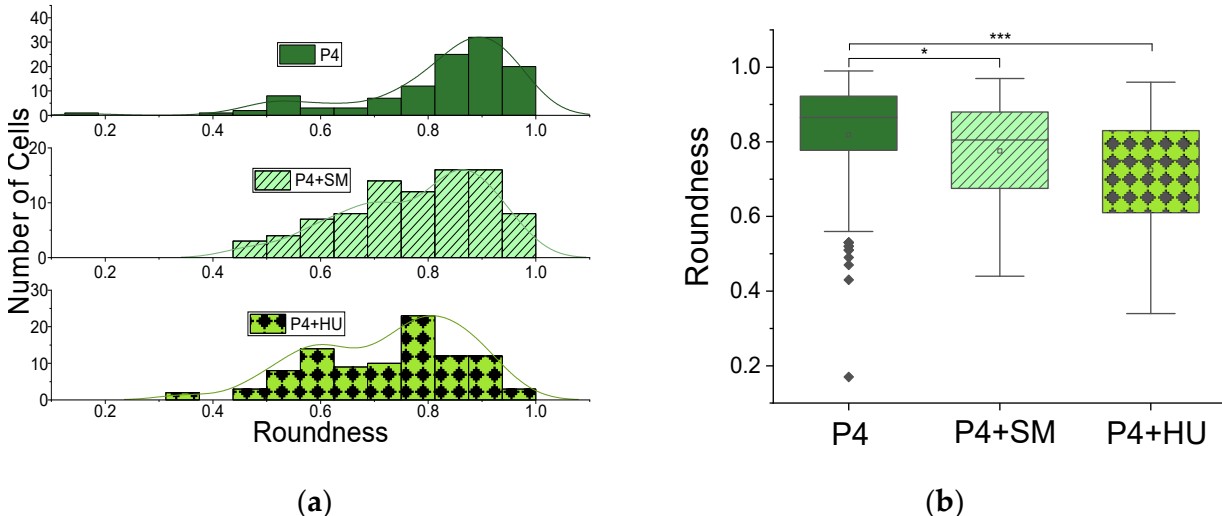

**Figure 4.** Morphometry of cells from MMM outlet: roundness. (**a**) Histogram of roundness (Equation (2)) and kernel smooth curves for RBCs from patient 4, showing morphological heterogeneity in untreated cells, P4; in cells treated with sodium metabisulfite, P4 + SM; and cells treated with hydroxyurea, P4 + HU. (**b**) Box plots for the roundness data in (**a**). Cell numbers (*n*) and roundness (*R*) for each condition are: P4, *n* = 114, *R* = 0.61 ± 0.02; P4 + SM, *n* = 88, *R* = 0.64 ± 0.02; and P4 + HU, *n* = 96, *R* = 0.50 ± 0.03. One-way ANOVA shows statistically significant (*** $p < 0.001$) differences between P4 and P4 + HU and (* $p < 0.05$) between P4 and P4 + SM.

Again, the heterogeneity of RBC morphology in the same patient is made clear in Figure 4a, where the histogram of steady-state roundness, P4, has two peaks highlighted by the kernel smooth curve. Unlike circularity (Figure 3 and Equation (1)), roundness does not account for roughness of edges since it does not use the perimeter in its definition or calculation. It depicts the entire geometry of the object (cell). Here, it confirms the heterogeneity found in the circularity results. The higher peak in the roundness distribution (Figure 4) corresponds to normal RBCs and the lower to sickled RBCs. The distribution changed when cells were treated with sodium metabisulfite (SM) to induce sickling, as seen in the P4 + SM result of Figure 4a. When the cells were created with hydroxyurea (HU), the distribution again departed from the steady-state situation as seen in the P4 + HU result of Figure 4a. Box plots for the roundness data are shown in Figure 4b with statistically significant differences found among P4, P4 + SM (* $p < 0.05$) and P4 + HU (* $p < 0.001$).

To quantify the morphological heterogeneity seen in RBCs of patient 4 in every condition measured, including steady state or normoxic (P4); induced hypoxic state, P4 + SM; and treatment state, P4 + HU, we measured the skewness and kurtosis of the circularity distributions in Figure 3 and the roundness distributions in Figure 4. Skewness measures the asymmetry of a distribution, and changes in skewness should enable the quantification of dynamic changes in proportions of sickled versus healthy RBCs in the same individual. Since kurtosis measures departure of a distribution from normality, it should enable the further quantification of heterogeneity based on outliers, even when homogeneity is approached. The mean circularity and mean roundness of the data in Figures 3 and 4 are tabulated in Table 1, along with the corresponding skewness and kurtosis for each condition.

**Table 1.** Summary of morphometric results.

|  | Skewness (@Circularity) | Kurtosis (@Circularity) | Circularity Equation (1) | Skewness (@Roundness) | Kurtosis (@Roundness) | Roundness Equation (2) |
|---|---|---|---|---|---|---|
| P4 | −0.64 | −0.98 | 0.61 ± 0.02 | −1.59906 | 2.79852 | 0.81 ± 0.01 |
| P4 + SM | −0.81 | −0.26 | 0.64 ± 0.02 | −0.59224 | −0.46862 | 0.78 ± 0.01 |
| P4 + HU | 0.17 | −1.54 | 0.50 ± 0.03 | −0.49392 | −0.46354 | 0.72 ± 0.01 |

### 3.2. Ex Vivo Active Monitoring of Patients' RBC

3.2.1. Monitoring of "VOC" via Induced Hypoxia

Using 2% sodium metabisulfite (SM) to induce hypoxia in the RBC samples, we found a significant decrease ($p < 0.001$) in transit time through the MMM for patient 1 following hypoxia (Figure 5), a significant increase ($p < 0.001$) in transit time through the MMM for patient 3 (Figure 6) and a non-significant change (at $p < 0.05$) in transit time through the MMM for patient 4 (Figure 7). These varied outcomes further underscore the need for patient-specific monitoring of rheological properties of RBCs to unravel their time-dependent contributions to VOC.

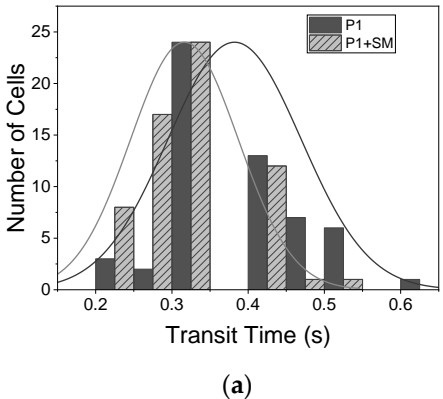
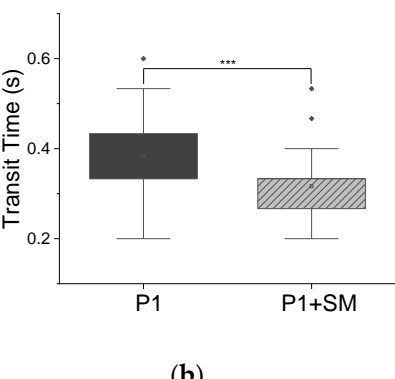

(**a**)                                                    (**b**)

**Figure 5.** Decrease in transit times following induced hypoxia. (**a**) Histogram distribution of transit times for patient 1, P1, normal (Gaussian) fit, with decrease in transit time when treated with SM. (**b**) Box plot of two-sample t-test, showing significant (*** $p < 0.001$) decrease in mean transit time *tt* from $0.38 \pm 0.01$ s to $0.32 \pm 0.01$ s for patient 1 (P1, $n = 56$ cells) and patient 1 treated with sodium metabisulfite (P1 + SM, $n = 63$).

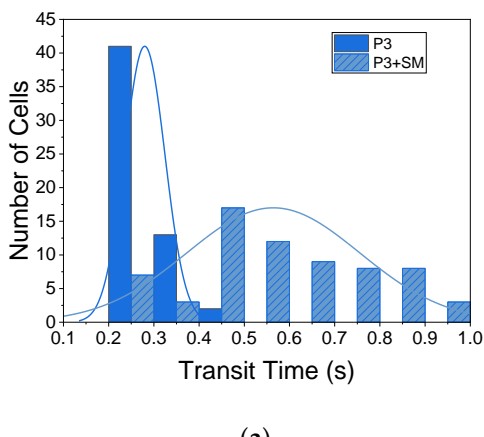
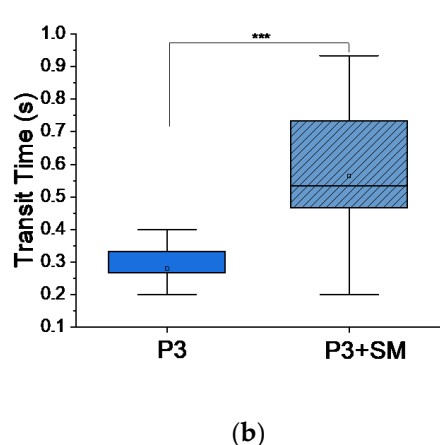

(**a**)                                                    (**b**)

**Figure 6.** Increase in transit times following induced hypoxia. (**a**) Histogram distribution of transit times for patient 3 (P3, $n = 56$), normal (Gaussian) fit, with increase in transit time when treated with SM (P3 + SM, $n = 67$). (**b**) Box plot of two-sample *t*-test, showing significant (*** $p < 0.001$) increase in mean transit time *tt* from $0.28 \pm 0.01$ s for P3 to $0.56 \pm 0.02$ s for P3 + SM.

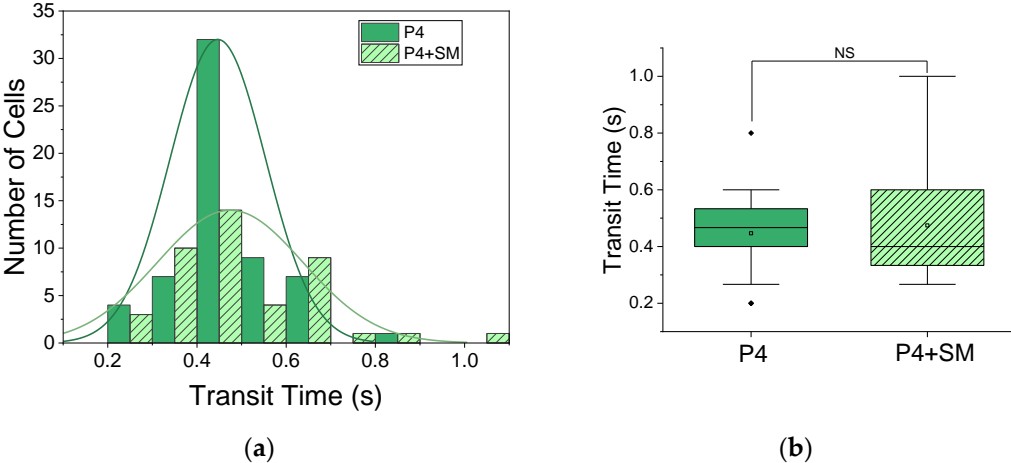

(**a**)   (**b**)

**Figure 7.** Non-significant change in transit time following induced hypoxia. (**a**) Histogram distribution for patient 4 (P4, *n* = 60) and patient 4 treated with SM (P4 + SM, *n* = 43). (**b**) Box plot of two-sample t-test, showing significant non-significant (NS) change in transit time from 0.45 ± 0.01 s for P4 to 0.47 ± 0.02 s for P4 + SM.

### 3.2.2. Monitoring of Drug Impact on 'VOC'

When we pre-incubated SCD patients' blood samples with 0.4 mg/mL of hydroxyurea to test if a therapeutic concentration of hydroxyurea has any effect on ex vivo-mimicked VOC using MMM, we found, contrary to our expectations, blockage of the MMM inlets after about 10 s of flow at all flow rates we applied: 0.9, 9 and 99 µL/h. The MMM inlet was blocked by long filamentous strands in P + HU conditions, strands absent in the non-treated samples. The inability to measure transit times accurately for HU-treated blood samples calls for multimodal microfluidic platforms that can provide morpho-rheological parameters without the constraints of capillary constrictions as obtained physiologically in larger vasculature.

### 4. Discussion

The main objective of this study was to show the feasibility of using an advanced lab-on-chip microfluidic platform to visualize, quantify and characterize properties of RBCs relevant to SCD complications, especially VOC. This objective was met, and for the few patients sampled, the presented results even agree with the previously known heterogeneity of SCD properties and also point to new directions for SCD research. Although we have presented results about the morpho-rheological properties of RBCs from SCD patients, we are careful not to generalize our findings owing to the limited number of patient samples. Interestingly, the MMM readouts for rheological (transit time) properties of the sRBCs showed significant variability ($p < 0.001$) among all patients tested. Changes in transit time between normoxic and hypoxic conditions ex vivo were diverse and patient-specific, including both increases and decreases in transit time between normoxic and hypoxic conditions. Furthermore, changes in transit time following ex vivo treatment with HU were also patient-specific. The morphological parameters (circularity and roundness) we measured suggest dynamic changes in the percentage of sickled RBCs in individual patients' blood, amplifying the call for personalized measurements using point-of-care technologies [23,31–33]. Combining these parameters during patient monitoring should enable the quantitative tracking of morphological parameters for correlation with clinical parameters, thereby potentially informing personalized clinical care. Our work shows the feasibility and potential utility of personalized clinical monitoring of SCD patients using an ex vivo real-time microfluidic lab-on-chip device even in resource-challenged settings, where they are most needed.

Recent computational models that use patient-specific data have also led to patient specificity in the morpho-rheological properties of RBCs of SCD patients even at the meso-

scopic level [35]. Xuejin et al. further suggest, based on their in silico work, that treatment with HU improves or worsens the rheological characteristics of blood in SCD depending on the degree of hypoxia [35]. Beyond SCD, Jung et al. have reviewed a vast body of evidence demonstrating the central role of microcirculation in inflammation, hyperviscosity, cell–cell interactions, endothelial dysfunction, tissue edema and hemodynamic regulation, with hypoxia being the leading feature in health and disease states [36]. Our morphometric results, which capture longitudinal heterogeneity within a specific patient as well as changes based on hypoxia or treatments, using shape descriptors such as roundness, circularity, skewness and kurtosis, will likely be useful in further unraveling components of VOC for SCD management as well as exploring the impact of hypoxia on cell components of the microcirculation in other hematological disorders. Our measurement of transit times for single RBCs using the MMM provides surrogate rheological characterization that also reveals heterogeneity within the same patient and between patients. This result agrees with Guruprasad et al.'s report of distinct biophysical RBC subpopulations with high inter-patient variability in SCD [23]. Our work illustrates the feasibility and potential utility of deploying the MMM along with high-throughput multimodal microfluidic platforms, such as the integrated automated particle tracking microfluidic [23] and real-time deformability cytometer [22], for patient-specific high-content cell analysis in studies involving patients with SCD and other blood disorders.

The feasibility of ex vivo active monitoring of RBCs of SCD patients is demonstrated in this work. The measurement of the effect of HU treatment and the impact of induced hypoxia on RBCs at specific time points were evaluated, demonstrating the capability of time-dependent monitoring. Our results show a significant increase, decrease and even non-significant change in transit time, which all fit with the dynamic nature of the changes induced by hypoxia and HU treatments of SCD patients. A recent multicenter study of hydroxyurea in SCD patients revealed increased deformability of RBCs for patients taking HU, measured using ektacytometry [37]. However, HU had other effects, such as changes in oxidative stress, nitrite levels and the nitric oxide synthase signaling pathway [30]. The blockage of the MMM inlet in the case of HU-treated RBC samples in our study opens the way for broader monitoring of the impact of HU treatment and other drugs as well as drug candidates on the circulatory system. The MMM method has the advantage of providing surrogate visualization of events as they occur in the microcirculation, and the measurements closely mimic actual events in human circulation. Therefore, it has the potential of increasing understanding and expediting research into the benefits or otherwise intervention of agents under research and drug development.

## 5. Conclusions

We have successfully deployed a cutting-edge lab-on-chip device for ex vivo monitoring of blood samples derived from SCD patients. Our work illustrates the feasibility of using the MMM for personalized care of SCD patients in a resource-challenged setting. Information from the ex vivo monitoring potentially elucidates VOC in individual patients, engendering more evidence-based clinical interventions. Our results suggest using the MMM and other advanced microfluidic platforms for bedside-to-bench tackling of questions in the clinical management of SCD and bench-to-bedside translation of emerging findings in SCD and other hematological research.

**Supplementary Materials:** The following supporting information can be downloaded at: https://www.mdpi.com/article/10.3390/app12094394/s1. SI Figure S1. Morphometry of RBC images from MMM outlet using Image J. SI Table S1: Anonymized patient data including sex, age and full blood counts; SI I: Consent Form; SI II: Questionnaire; SI III: Inclusion/Exclusion Criteria; SI IV: Ethics Committee Approval.

**Author Contributions:** Conceptualization, M.I.A., A.O. and A.E.; Data curation, M.I.A.; Formal analysis, M.I.A.; Funding acquisition, A.E.; Investigation, E.E., S.A., S.-P.B. and C.N.; Methodology, M.I.A., E.E., O.G., A.O., N.T., S.A., N.Z., C.N., W.E., J.G. and A.E.; Project administration, A.E.;

Resources, J.G. and A.E.; Software, A.E.; Supervision, M.I.A. and A.E.; Validation, A.E. and M.I.A.; Writing—original draft, M.I.A. and A.E.; Writing—review and editing, N.T. All authors have read and agreed to the published version of the manuscript.

**Funding:** This research received no external funding. Major equipment (MMM, microscopes, syringe pump) were provided by AE's laboratory at Creighton University. MIA provided reagents for experiments (HU, SM) and refreshments for participants. The Joseph Ukpo Hospitals and Research Institutes, JUHRI, provided partial funding from donations made through Friends of JUHRI, a 501(c)3 non-profit organization and public charity. JUHRI provides 100% free healthcare to the rural poor in Akwa Ibom and Cross River States of Nigeria.

**Institutional Review Board Statement:** The study was conducted according to the guidelines of the Declaration of Helsinki and approved by the Institutional Review Board (Health Research Ethics Committee) of the University of Calabar Teaching Hospital, under the registration number NHREC/07/10/2012, and was assigned the following protocol code for study: UCTH/HREC/33/498, following approval on 12 August 2016 (see SI IV for approval letter).

**Informed Consent Statement:** Informed consent was obtained from all subjects involved in the study (see SI I: Consent Form).

**Data Availability Statement:** The data presented in this study are available upon reasonable re-quest from the corresponding author and will only be transmitted in an anonymized form. The data are not publicly available due to patient confidentiality.

**Acknowledgments:** Authors received administrative, technical and material support from the Hematology Department of the University of Calabar Teaching Hospital, including routine laboratory reagents such as EDTA.

**Conflicts of Interest:** The authors declare no conflict of interest.

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
