# Peer review of "Microfluidic Microcirculation Mimetic as a Tool for the Study of Rheological Characteristics of Red Blood Cells in Patients with Sickle Cell Anemia"

_applsci, doi:10.3390/app12094394_

Round 1

Reviewer 1 Report

The article presented is an impressive collaboration of scientific and clinical researchers that aim to address clinical questions regarding the management of sickle cell disease, particularly vaso-occlusive crisis. The team uses their device in regions of the world where the burden of sickle cell disease is the highest and where resources can be limited. The team addresses important features of the disease such as the rheological profile of sickle red blood cells and aims to understand features of the disease that contribute to VOC as well as the heterogeneity seen in the patient’s clinical profile. Overall, the study has much potential and would benefit from a closer look into the experimental methods and the interpretation of the results with respect to the morphological features of normal and sickled red blood cells.

 Strengths: The transit time data has the strongest technical support and yields interesting results. The transit time readout shows inter-patient variability for patients with SCA which has significant importance when considering the clinical presentation of the disease and the corresponding variability seen in the patient population. The transit time analysis and morphology analysis are performed on the same patient which strengthens the potential impact of this platform. The study also claims an impressive patient sample size (n = 25).

Weaknesses: This study would benefit from a closer look into the data acquisition and morphology readouts as the interpretation of circularity and roundness measurements is not complete. The study claimed a sample size of 25 patients (section 3.1.1) but only reported data for 4 patients. Additionally, the morphology analysis was only performed on 1 patient. An explanation for this is needed. Otherwise, the implication is that the data were cherry-picked to support the authors’ conclusions. The study also needs control data (healthy donor blood) for transit time and morphology readouts. For future work, it would also be interesting to include the patients hematological and clinical profile. For instance, are patients typically on drugs like hydroxyurea and are they regularly transfused?

Specific concerns to be addressed:

  • Please include supplementary information which is referenced in the main text.
  • Device motivation and state of the art:
    • What are the advantages of this device over others? For example, other devices exist that measure deformability and transit time. How does this compare (i.e. does it amplify the signal, is it a more repeatable measurement, is it simple and easy to use)?
      • Tomaiuolo, M. Simeone, V. Martinelli, B. Rotoli and S. Guido, Soft Matter, 2009, 5, 3736–3740.
      • Alapan Y, Matsuyama Y, Little JA, Gurkan UA. Dynamic deformability of sickle red blood cells in microphysiological flow. Technology (Singap World Sci). 2016;4(2):71-79. doi:10.1142/S2339547816400045
      • Adamo A, Sharei A, Adamo L, Lee B, Mao S, Jensen KF. Microfluidics-based assessment of cell deformability. Anal Chem. 2012;84(15):6438-6443. doi:10.1021/ac300264v
    • The methods state there were three generations of device with constriction widths of 15 um, 7 um, and 5 um. What device was used for the data presented?
  • Data acquisition:
    •  Please clarify the region of interest for data acquisition of the circularity and roundness measurements in Figure 1. The text states in the methods “We have used several shape descriptors to parameterize and quantify the morphology of red blood cells in the inlet and outlet of the MMM.” but the results section claims measurements were only taken at the outlet.
    • Are transit time and circularity/roundness independent measurements?
    • How does the circularity measurement account for cell rotation about its axis? For example, would an un-sickled red blood cell (discocyte, asymmetric) have a different readout if it was rotated onto its side? Does the device have a flow focusing feature at the outlet to ensure the primary axis of the cell is parallel to the bottom of the channel?
      • Red blood cell dynamics in flow: S. Atwell, C. Badens, A. Charrier, E. Helfer and A. Viallat, Physiol.,2022, 12
      • Goldsmith H. L., Marlow Jean and MacIntosh Frank Campbell 1972 Flow behaviour of erythrocytes - I. Rotation and deformation in dilute suspensionsProc. R. Soc. Lond. B.182351–384
    • Can you include example images of your red blood cells at different degrees of circularity and roundness? “for instance, cells 9 and 14 in panel 6 (Results) are clearly normal and sickled, respectively, as in the original image (panel 1) and have C= 0.92±0.02 and C= 0.79±0.02, R = 0.95±0.02 and R = 0.64±0.01.” This histogram plots also report circularities of < 0.5. What do these cells look like?
    • How is the transit time data computed from the images?
  • Control data missing:
    • Data set would benefit from analysis of healthy blood samples as a control. This will aid in the interpretation of results for diseased blood (HbSS RBC), HbSS RBC + SMS and HbSS RBC + HU as well as strengthen the claim regarding patient to patient variability for SCD under the hypothesis that red blood cells from healthy donors would show less variability.
  • Morphological interpretation:
    • Please support the use of circularity and roundness as morphological classifiers for sickled and healthy red blood cells in unrestricted flow conditions (device outlet). The abstract states “The MMM enabled a patient-specific quantification of shape de- scriptors (circularity and roundness) and transit time through the capillary constrictions, which are readouts for morphorheological properties implicated in VOC.”
    • Please clarify which peak corresponds to each cell morphology in Figure 3. The text states “the histogram of steady-state circularity, P4, has two peaks as made clearer by the kernel smooth curve. The higher peak corresponds to normal RBCs and the lower, to sickled RBCs, with a broad range that shows varying degrees of sickling. The distribution is altered when cells are treated with sodium metabisulfite (SM), to induce sickling, as seen in the P4+SM result of Figure 3 (a).” Assuming the high peak is closer to circularity = 1 and the low peak is closer to circularity = 0, please explain why in figure 3 there is a right shifted effect to the “higher” peak for P4+SM (closer to circularity = 1). Why would there be less sickle cells if deoxygenation from sodium metabisulfite induces sickling? Please explain why there is a left shift in the distribution to the lower peak with P4 + HU. Why would there be more sickle cells at ambient oxygen partial pressures if hydroxyurea has known therapeutic effects?
    • The roundness measurement results are slightly more supported except for the P4 + HU condition. Why do the red blood cells become less round when treated with hydroxyurea? Are there morphological effects due to incubating the red blood cells
    • Morphology classification has been performed before in both static and flow conditions. How do these results compare with the circularity and roundness measurement? If they are different, why? What are the advantages of this device for quantifying morphology?
      • Suriany S, Xu I, Liu H, et al. Individual red blood cell nitric oxide production in sickle cell anemia: Nitric oxide production is increased and sickle shaped cells have unique morphologic change compared to discoid cells. Free Radic Biol Med. 2021;171:143-155
      • Wheeless, L.L., Robinson, R.D., Lapets, O.P., Cox, C., Rubio, A., Weintraub, M. and Benjamin, L.J. (1994), Classification of red blood cells as normal, sickle, or other abnormal, using a single image analysis feature. Cytometry, 17: 159-166. https://doi.org/10.1002/cyto.990170208
      • Safca, D. Popescu, L. Ichim, H. Elkhatib and O. Chenaru, "Image Processing Techniques to Identify Red Blood Cells," 2018 22nd International Conference on System Theory, Control and Computing (ICSTCC), 2018, pp. 93-98, doi: 10.1109/ICSTCC.2018.8540708.
    • Another point, RBCs have different morphologies under different shear stresses. What is the typical morphology (discocyte, parachute, slipper) for the imposed shear stress?
      • Reichel F, Mauer J, Nawaz AA, Gompper G, Guck J, Fedosov DA. High-Throughput Microfluidic Characterization of Erythrocyte Shapes and Mechanical Variability. Biophys J. 2019;117(1):14-24. doi:10.1016/j.bpj.2019.05.022
  • HU treatment:
    • Please add a reference that supports the methods for HU treatment in this study and the bioactivity of hydroxyurea in blood ex – vivo.
  • Samples reported:
    • The authors report measuring 25 individual blood samples, but only 4 patients are included in the results. This discrepancy must be clarified. 

Author Response

Dear Reviewer, Please see the attachment  for our point -by-point response to your valuable comments. 

Dr Marcus Inyama Asuquo

Reviewer 2 Report

Thank you for giving me the opportunity to review the paper by Asuquo et al. The authors investigated plasma samples from patients with sickle cell anaemia using a lab-on-chip microfluidic platform.

The topic is interesting, but the paper needs some fundamental redesign.

  • The abstract gives a lot of information about the Sickle cell disorder in general, but some basic information about the study itself are missing (samples size, included patients). Furthermore, no results are presented, just a conclusion. This is not acceptable for a scientific paper.
  • The introduction is informative, but one essential point is missing for me: what is the state of the art diagnostic/therapy up to now? Are there any other devices available and what are differences between the systems?
  • Methods: I cannot find any information about the patients that were enrolled. What are the clinical characteristics? And how many plasma samples were taken from which patient? Have there been special time intervals? Etc….
  • Please avoid any interpretations in the results, this should be done only in the discussion. Furthermore, it is difficult to follow, what was really done in the experiments. But I think this is also a problem of the insufficient methods section. So please give both sections more structure and clarity!

Author Response

Dear Reviewer, Please see the attachment for a point-by-point response to your valuable comments.

Dr Marcus Inyama Asuquo

Round 2

Reviewer 1 Report

The authors addressed the concerns from the review.

Author Response

Dear Editor, Please, see the attachment in the box for our response to comments by reviewer 1. in the second round.

Marcus Inyama Asuquo

Reviewer 2 Report

Honestly, I am a little bit disappointed about the response of the authors. There are still substantial things missing.

1.) Abstract: Basic informations about the patients are still missing and the abstract is lacking of any results. Again: This is not acceptable for a scientific paper.

2.) A table with all patient characteristics (e.g. age, ASA status, co-morbidities, etc.) is still missing and an undifferentiated upload of study sheets/protocols is not a suffcient compensation. By the way, these information should be part of the paper and not the supplement.

Just in case, the information are still missing in the next version, I will reject the paper.

Author Response

Dear Editor, Please, see the attachment in the box for our response to reviewer 2 comments in the second round.

Marcus Inyama Asuquo
